# Dose Consideration of Lenvatinib’s Anti-Cancer Effect on Hepatocellular Carcinoma and the Potential Benefit of Combined Colchicine Therapy

**DOI:** 10.3390/cancers15205097

**Published:** 2023-10-22

**Authors:** Zu-Yau Lin, Ming-Lun Yeh, Po-Cheng Liang, Po-Yao Hsu, Chung-Feng Huang, Jee-Fu Huang, Chia-Yen Dai, Ming-Lung Yu, Wan-Long Chuang

**Affiliations:** 1Division of Hepatobiliary Medicine, Department of Internal Medicine, Kaohsiung Medical University Hospital, Kaohsiung 80756, Taiwan; yeh_ming_lun@yahoo.com.tw (M.-L.Y.); pocheng.liang@gmail.com (P.-C.L.); u9501067@gmail.com (P.-Y.H.); fengcheerup@gmail.com (C.-F.H.); jfliver@kmu.edu.tw (J.-F.H.); daichiayen@gmail.com (C.-Y.D.); fish6069@gmail.com (M.-L.Y.); waloch@kmu.edu.tw (W.-L.C.); 2Department of Internal Medicine, Faculty of Medicine, College of Medicine, Kaohsiung Medical University, Kaohsiung 80708, Taiwan; 3Center for Cancer Research, Kaohsiung Medical University, Kaohsiung 80708, Taiwan; 4Center for Liquid Biopsy and Cohort Research, Kaohsiung Medical University, Kaohsiung 80708, Taiwan; 5School of Medicine, College of Medicine, National Sun Yat-sen University, Kaohsiung 804201, Taiwan; 6School of Medicine and Doctoral Program of Clinical and Experimental Medicine, College of Medicine, Center of Excellence for Metabolic Associated Fatty Liver Disease, National Sun Yat-sen University, Kaohsiung 804201, Taiwan

**Keywords:** colchicine, dose-dependent effect, hepatocellular carcinoma, lenvatinib, multikinase inhibitor, *NANOG*

## Abstract

**Simple Summary:**

The long-term benefits of lenvatinib on the treatment of advanced hepatocellular carcinoma (HCC) are still unsatisfactory. The search for a new drug to promote lenvatinib’s anti-cancer effect is an urgent issue. Whether the response of HCC to lenvatinib is dose-dependent also still needs to be clarified. The aims of this study were to investigate the dose-dependent anti-cancer effect of lenvatinib on HCC cells and the potential benefit of combined colchicine therapy. Four primary cultured HCC cell lines were applied for experiments. Combined analysis of the results of differential expressions of the genes (11 lenvatinib target genes and *NANOG*) and the anti-proliferative effect indicated that the anti-cancer effect of lenvatinib on HCC was not dose dependent. Combined clinically achievable plasma colchicine concentration with lenvatinib can promote the total anti-cancer effects on HCC.

**Abstract:**

Purpose: The dose-dependent anti-cancer effect of lenvatinib on hepatocellular carcinoma (HCC) cells and the potential benefit of combined colchicine therapy were investigated. Methods: Four primary cultured HCC (S103, S143, S160, S176) cell lines were investigated by differential expressions of genes (11 lenvatinib target genes and *NANOG*) and anti-proliferative effect using clinically achievable plasma lenvatinib (250, 350 ng/mL) and colchicine (4 ng/mL) concentrations. Results: Colchicine showed an anti-proliferative effect on all cell lines. Lenvatinib at 250 ng/mL inhibited proliferation in all cell lines, but 350 ng/mL inhibited only three cell lines. For lenvatinib target genes, colchicine down-regulated more genes and up-regulated less genes than lenvatinib did in three cell lines. Lenvatinib up-regulated *NANOG* in all cell lines. Colchicine down-regulated *NANOG* in three cell lines but up-regulated *NANOG* with less magnitude than lenvatinib did in S103. Overall, combined colchicine and 250 ng/mL lenvatinib had the best anti-cancer effects in S143, with similar effects with combined colchicine and 350 ng/mL lenvatinib in S176 but less effects than combined colchicine and 350 ng/mL lenvatinib in S103 and S160. Conclusions: Lenvatinib does not show a dose-dependent anti-cancer effect on HCC. Combined colchicine and lenvatinib can promote the total anti-cancer effects on HCC.

## 1. Introduction

Lenvatinib is an approved first-line oral multikinase molecular inhibitor for the treatment of advanced hepatocellular carcinoma (HCC) [1]. The anti-cancer mechanisms of lenvatinib include suppression of vascular endothelial growth factor receptor (VEGFR) 1–3, fibroblast growth factor receptor (FGFR) 1–4, platelet-derived growth factor receptor (PDGFR) α, as well as proto-oncogenes RET and KIT expressions [2,3,4]. In comparison with sorafenib, lenvatinib was shown to be non-inferior to sorafenib in overall survival and even showed much better median progression-free survival and a higher objective response rate than sorafenib [5,6,7,8]. However, the long-term benefits of lenvatinib on the treatment of advanced HCC are still unsatisfactory. Combined lenvatinib with an immunotherapeutic agent might have the possibility to obtain better anti-cancer activity than lenvatinib or immunotherapeutic agent alone, but the overall toxicities and the cost-effect still need to be considered [9]. Therefore, the search for a new drug to enhance the lenvatinib effects on HCC is a clinically urgent issue. On the other hand, increasing lenvatinib plasma concentration usually increases its side effects [10,11,12]. Whether the response of HCC to lenvatinib is dose-dependent still needs to be clarified.

Colchicine is a widely applied and very cheap tricyclic alkaloid which has been shown to have dose-dependent anti-cancer effect within its clinically achievable plasma concentrations on primary cultured HCC cells and cancer-associated fibroblasts [13]. This drug has been shown to have the potential for the palliative treatment of advanced HCC [14]. The purpose of this study was to investigate the dose-dependent anti-cancer effect of lenvatinib on HCC and the potential benefit of colchicine in combined therapy evaluated by differential expressions of target genes and an anti-proliferative assay. *NANOG* is a cancer stem cell marker which has been shown as a hazard factor to predict poor prognosis in patients with HCC [15]. Our previous experiments using two other multikinase molecular inhibitors including sorafenib and regorafenib in primary cultured HCC cells showed that these drugs could up-regulate *NANOG* within their therapeutic ranges [16]. Therefore, *NANOG* was selected as one of the target genes. Eleven genes including *FGFR1*, *FGFR2*, *FGFR3*, *FGFR4*, *FLT1* (*VEGFR1*), *FLT4* (*VEGFR3*), *KDR* (*VEGFR2*), *KIT*, *PDGFRA*, *PDGFRB*, *RET* were selected as lenvatinib target genes based on the previous reports [2,3,4]. For the best possibility to represent the situation in patients, clinically feasible colchicine (4 ng/mL [17,18,19] and lenvatinib 250 ng/mL and 350 ng/mL, representing low and high therapeutic concentrations) [20] concentrations were selected for the experiments. Moreover, primary cultured HCC cells at their 4th–6th passage were applied for the investigation to maintain the characters of heterogenicity of the cancer cells within the tumor. Gene names were based on the official symbols from the HUGO Gene Nomenclature Committee.

## 2. Materials and Methods

### 2.1. Cell Lines and Drugs

Four primary cultured HCC cell lines (S103, S143, S160, S176) established by our institution were applied [13,16]. All patients had chronic hepatitis B and liver cirrhosis. The TNM tumor staging [21] for four patients was stage IIIB in two (S103, S143), stage IVA in one (S160), and stage IVB in the remaining one (S176). The serum alpha-fetoprotein levels in three patients (S103, S160, S176) were larger than 1400 ng/mL, and the remaining one (S143) was 24.45 ng/mL (normal range < 20 ng/mL). Cells were cultured with 10% fetal bovine serum, 90% DME/HIGH glucose, supplemented with 20 mM of L-glutamine, 100 units/mL of penicillin, and 100 μg/mL of streptomycin (HyClone, Logan, UT, USA) in 37 °C and a humidified atmosphere of 5% CO_2_ and 95% air. Lenvatinib purchased from MedChem Express (Monmouth Junction, NJ, USA) was dissolved in dimethyl sulfoxide (DMSO) to give 250 ng/mL or 350 ng/mL with a DMSO concentration of 1% (*v*/*v*) in the culture. Colchicine was purchased from Sigma Aldrich (St. Louis, MO, USA). DMSO was added to the colchicine alone group and control group with a final DMSO concentration of 1% (*v*/*v*) in the culture. This study was approved by the Institutional Review Board (Kaohsiung Medical University Chung-Ho Memorial Hospital Institutional Review Board-I, KMUHIRB-GI-20180036). Informed consents were obtained from all patients for the collection of cancer cells for primary cultures.

### 2.2. Anti-Proliferative Experiments

Cancer cells were incubated in 96-well culture plates with a serum-containing medium for 48 h. The culture medium was changed to a serum-free medium with 4 ng/mL of colchicine, 250 ng/mL of lenvatinib, 350 ng/mL of lenvatinib, 4 ng/mL of colchicine + 250 ng/mL of lenvatinib, and 4 ng/mL of colchicine + 350 ng/mL of lenvatinib or without any drug. The proliferation assay was performed after incubation for a further 72 h using the premixed WST-1 cell proliferation reagent (Clontech Laboratories, Inc., A Takara Bio Company, Mountain View, CA, USA) detected using an automated microplate reader (Synergy H1 hybrid multi-mode reader with Gen5 software, BioTek Instruments, Inc., Winooski, VT, USA) with an absorbance of 450 nm wavelength (reference wavelength 630 nm). The results from 16 replicated wells were applied for statistical calculation.

### 2.3. Quantitative Reverse Transcriptase-Polymerase Chain Reaction (qRT-PCR) Experiments

Cancer cells were seeded in 25 cm^2^ plastic flasks with a serum-containing medium. When the growth of the cells was over 80% of the total growth area, the medium was changed to a serum-free medium with 4 ng/mL of colchicine, 250 ng/mL of lenvatinib, 350 ng/mL of lenvatinib, 4 ng/mL of colchicine + 250 ng/mL lenvatinib, and 4 ng/mL of colchicine + 350 ng/mL of lenvatinib or without any drug for a further 24 h for the qRT-PCR experiments. The procedures for qRT-PCR were the same as with previous studies [13,16]. The mean of triple qRT-PCR determinations for each gene was calculated for analysis. The up-regulation of genes was defined as a gene expression fold change >1.3, and the down-regulation of genes was defined as a gene expression fold change <0.7 [13,16]. Eleven lenvatinib target genes and *NANOG* were studied. The reference gene was the housekeeping gene *TBP* (TATA box binding protein). The PCR primers used were 5′-AGAATATCATCAACCTGCTGGG-3′ sense primer and 5′-TTGGAGGCATACTCCACGAT-3′ anti-sense primer for *FGFR1*, 5′-CAGAATGGATAAGCCAGCCA-3′ sense primer and 5′-GCTTGAACGTTGGTCTCTGG-3′ anti-sense primer for *FGFR2*, 5′-GCCTCCTCGGAGTCCTTG-3′ sense primer and 5′-AAGACCAACTGCTCCTGCTG-3′ anti-sense primer for *FGFR3*, 5′-GCTGCTTTGGCCAGGTAGTA-3′ sense primer and 5′-AGGTCCTTGTCAGAGGCGTT-3′ anti-sense primer for *FGFR4*, 5′-AATGCCACCTCCATGTTTGA-3′ sense primer and 5′-GGTTTGCTGTCAGTCCAGGT-3′ anti-sense primer for *FLT1*, 5′-CCACGCACCAGACGCTTG-3′ sense primer and 5′-GGACGACGAAGATGACCTTATACG-3′ anti-sense primer for *FLT4*, 5′-TCTTGCCTCAGAAGAGCTGAA-3′ sense primer and 5′-GCCTTCAGATGCCACAGACT-3′ anti-sense primer for *KDR*, 5′-GCAGATTTCAGAGAGCACCAA-3′ sense primer and 5′-ATTGATCCGCACAGAATGGT-3′ anti-sense primer for *KIT*, 5′-TCTCGTATTTGCTGCATCGT-3′ sense primer and 5′-CACTCGGTGAAATCAGGGTAA-3′ anti-sense primer for *NANOG*, 5′-GACATTGACCCTGTCCCTGA-3′ sense primer and 5′-AACCCGTCTCAATGGCACT-3′ anti-sense primer for *PDGFRA*, 5′-CCTTACCACATCCGCTCCATC-3′ sense primer and 5′-TCACACTCTCCGTCACATTGC-3′ anti-sense primer for *PDGFRB*, 5′-CACCGCTGGTGGACTGTAAT-3′ sense primer and 5′-GGACTCTCTCCAGGCCAGTT-3′ anti-sense primer for *RET*, and 5′-CAATTTAGTAGTTATGAGCCAGAG-3′ sense primer and 5′-TTCTGCTCTGACTTTAGCAC-3′ anti-sense primer for *TBP*.

### 2.4. Statistical Analysis

An unpaired two-tailed *t*-test was applied for calculating the significant difference between the two means. The *p* value < 0.05 was defined as statistically significant.

## 3. Results

### 3.1. Anti-Proliferative Experiments

Figure 1 showed the effects of colchicine and lenvatinib on proliferation. Colchicine had significant anti-proliferative effects on all cell lines. For lenvatinib, 250 ng/mL had anti-proliferative effects on all cell lines, but 350 ng/mL only showed anti-proliferative effects on three (S103, S143, S160) out of four cell lines. Lenvatinib at a concentration of 250 ng/mL showed similar anti-proliferative effect with 350 ng/mL in S103, which was a stronger anti-proliferative effect than 350 ng/mL in S143 but a weaker anti-proliferative effect than 350 ng/mL in S160. Adding colchicine to 250 ng/mL of lenvatinib showed similar anti-proliferative effects with 250 ng/mL of lenvatinib alone in S103 and S143, which was a stronger anti-proliferative effect than 250 ng/mL of lenvatinib alone in S160 but a weaker anti-proliferative effect than 250 ng/mL of lenvatinib alone in S176. Adding colchicine to 350 ng/mL of lenvatinib showed stronger anti-proliferative effects than 350 ng/mL lenvatinib alone in S103, S143, and S160. Although 350 ng/mL of lenvatinib showed no effect on S176 proliferation, combined 350 ng/mL of lenvatinib and colchicine could significantly inhibit proliferation.

### 3.2. qRT-PCR Experiments

#### 3.2.1. Influence on Expressions of Lenvatinib Target Genes

The results are shown in Table 1 and Table 2. In comparison with the effects of 250 ng/mL and 350 ng/mL of lenvatinib, colchicine induced more down-regulated genes and a smaller number of up-regulated genes in three cell lines (S103, S143, S160). For S176, colchicine induced more up-regulated genes than either 250 ng/mL or 350 ng/mL of lenvatinib did. Lenvatinib at a concentration of 350 ng/mL induced fewer up-regulated genes than 250 ng/mL did in S143 but one more up-regulated gene than 250 ng/mL did in S103. Although 350 ng/mL lenvatinib induced more down-regulated genes than 250 ng/mL did in S143 and S160, this concentration reversely reduced two down-regulated genes in comparison with 250 ng/mL in S176. Combined colchicine and 250 ng/mL of lenvatinib induced more down-regulated genes with a smaller number of up-regulated genes than 250 ng/mL of lenvatinib did in S103, S143, and S160. For S176, this combination caused two more up-regulated genes than 250 ng/mL of lenvatinib did. Combined colchicine and 350 ng/mL of lenvatinib induced more down-regulated genes than 350 ng/mL of lenvatinib did in all cell lines. Although this combination induced a smaller number of up-regulated genes than 350 ng/mL of lenvatinib did in S103 and S160, it reversely induced more up-regulated genes in comparison with 350 ng/mL of lenvatinib in S143 and S176.

#### 3.2.2. Influence on Expression of NANOG

The results are shown in Table 1. Colchicine induced down-regulation of *NANOG* in S143, S160, and S176 but up-regulation of *NANOG* in S103. The magnitude of up-regulated *NANOG* induced by colchicine in S103 was smaller than either 250 ng/mL or 350 ng/mL of lenvatinib. Lenvatinib at a concentration of either 250 ng/mL or 350 ng/mL induced up-regulation of *NANOG* in S103, S143, and S160. Lenvatinib at a concentration of 250 ng/mL also induced up-regulation of *NANOG* in S176. Combined colchicine with 250 ng/mL or 350 ng/mL of lenvatinib induced a smaller magnitude of up-regulated *NANOG* in comparison with both lenvatinib and colchicine alone in S103. Combined colchicine and 250 ng/mL of lenvatinib could obliterate the up-regulated *NANOG* in S143 and reverse the up-regulated *NANOG* in S160 and S176 to down-regulated as compared with 250 ng/mL of lenvatinib alone. In comparison with 350 ng/mL of lenvatinib alone, combined colchicine and 350 ng/mL of lenvatinib could decrease the magnitude of up-regulated *NANOG* in S143, reverse *NANOG* from up-regulation to down-regulation in S160, and cause down-regulation of *NANOG* in S176.

### 3.3. Combined Analysis the Results of Anti-Proliferative Effects and qRT-PCR Experiments

Table 3 shows the summary of the total experimental results. Combined colchicine with 350 ng/mL of lenvatinib had the best anti-cancer effects followed by combined colchicine with 250 ng/mL of lenvatinib in S103 and S160. For S143, combined colchicine and 250 ng/mL of lenvatinib showed the best anti-cancer effects. Combined colchicine and 250 ng/mL or 350 ng/mL of lenvatinib showed similar anti-cancer effects on S176.

## 4. Discussion

The dose-dependent anti-cancer effect is a well-known concept to describe that the efficiency of an anti-cancer drug will increase with an increased dose within its therapeutic range. This concept is widely adopted in the clinical prescription of cytotoxic agents or in the experiment for searching for a new agent with direct cytotoxicity. However, the present results showed that lenvatinib were not consistent with this concept. The explanation was that targeting of particular genes related to angiogenesis and proliferation rather than direct cytotoxicity were the major anti-cancer mechanisms for multikinase molecular inhibitors. High lenvatinib concentration (350 ng/mL) caused more up-regulated lenvatinib target genes in S103 and S176 and a larger degree of up-regulated *NANOG* in S143 and S160 than low lenvatinib concentration (250 ng/mL). Moreover, 350 ng/mL of lenvatinib had no anti-proliferative effect on S176. These findings were in accordance with our previous studies on two other multikinase molecular inhibitors (sorafenib and regorafenib) [16]. On the other hand, the composition of tumors in HCC is heterogeneity containing cancer stem cells and cancer cells of different characteristics [22,23,24,25]. Since the intra-cellular interaction of genes and signaling pathways are determined by the characteristics of the cancer cells, the anti-cancer effects through targeting on particular genes can, thus, be quite variable among different cancer cells. On the other hand, the present study also demonstrated that lenvatinib could up-regulate *NANOG* in S103, S143, and S160 at a concentration of either 250 ng/mL or 350 ng/mL and in S176 at a concentration of 250 ng/mL. These results were in consistent with the effects of sorafenib and regorafenib on these cell lines [16]. Cancer cells with *NANOG* expression are considered cancer stem cells, also known as tumor-initiating cells or cancer cells with stem cell-like properties. These cells exhibit an enhanced ability of self-renewal, clonogenicity, initiation of tumors, and resistance to therapeutic agents [26,27,28]. Meta-analysis also showed that patients with a positive *NANOG* expression of cancer cells in an HCC tumor had poor 3-year and 5-year overall survival and disease-free survival rate [15]. The above results in combination with a previous report [16] indicate the essential therapeutic limitation for multikinase inhibitors in the treatment of HCC. Moreover, these drugs may also have the possibility to up-regulate genes favored for tumor progression particularly for the *NANOG*. Therefore, combined multikinase inhibitors with other drugs with different anti-cancer mechanisms is a reasonable choice in the treatment of patients with advanced HCC.

Tubulin protein plays an essential role in cell division and intracellular transportation. The inhibition of microtubule formation by targeting the tubulin protein can induce cell death by apoptosis. An efficient cytotoxic tubulin inhibitor is a credible solution for treating many species of cancers [29]. Colchicine is a microtubule destabilizer which has very strong binding capacity to tubulin to perturb the assembly dynamics of microtubules [30,31] and also can increase cellular-free tubulin to limit mitochondrial metabolism in cancer cells [32]. Differences in the characteristics of cancer cells have no remarkable influence on the anti-proliferative effect of colchicine [13]. Besides the direct cytotoxic effect, the present results also show that colchicine could down-regulated more lenvatinib target genes and decrease the number of up-regulated target genes as compared with lenvatinib in three cell lines. Colchicine also caused down-regulation of *NANOG* in three cell lines and reduced the degree of up-regulated *NANOG* caused by either 250 ng/mL or 350 ng/mL of lenvatinib in the remaining one cell line (S103). Combined colchicine with lenvatinib demonstrated better anti-cancer effects than either colchicine or lenvatinib alone in view of an anti-proliferative effect and the expressions of *NANOG* and lenvatinib target genes. Since colchicine is very cheap and the side effects using our novel colchicine dosage schedule were clinically acceptable [14], this drug can be considered to be applied in combination with lenvatinib for the palliative treatment of advanced HCC.

The limitation for the present study was that the qRT-PCR results did not receive further confirmation by an immunoblotting assay. Since different characteristics of cancer cells determine the interaction of genes and signaling pathways and the composition of HCC tumor is heterogeneity, the actual clinical significance for those up-regulated genes induced by colchicine, Lenvatinib, or their combinations still need to be further investigated. On the other hand, the present study was unable to elucidate the definite relation between cellular characteristics and lenvatinib effects on HCC due to only four cell lines being studied. Further large-scale, extensive experiments are required to clarify this important topic. Nevertheless, the present results can provide new insight for understanding the effects of lenvatinib on HCC and the potential role of colchicine on combined therapy. Animal experimentation was not considered in the present study. The key reason was that no proven lenvatinib doses and treatment durations could be applied in animal experiments to actually reflect human response and, thus, could be misleading in the interpretation of the results.

## 5. Conclusions

Lenvatinib does not show a dose-dependent anti-cancer effect on HCC. With the combined consideration of lenvatinib side effects and our results, starting from low rather than high therapeutic doses and gradually increasing the dose in case of no obvious evidence of anti-cancer effects might be a suitable way to prescribe this drug. Combined colchicine with lenvatinib can promote the total anti-cancer effects on HCC.

## Figures and Tables

**Figure 1 cancers-15-05097-f001:**
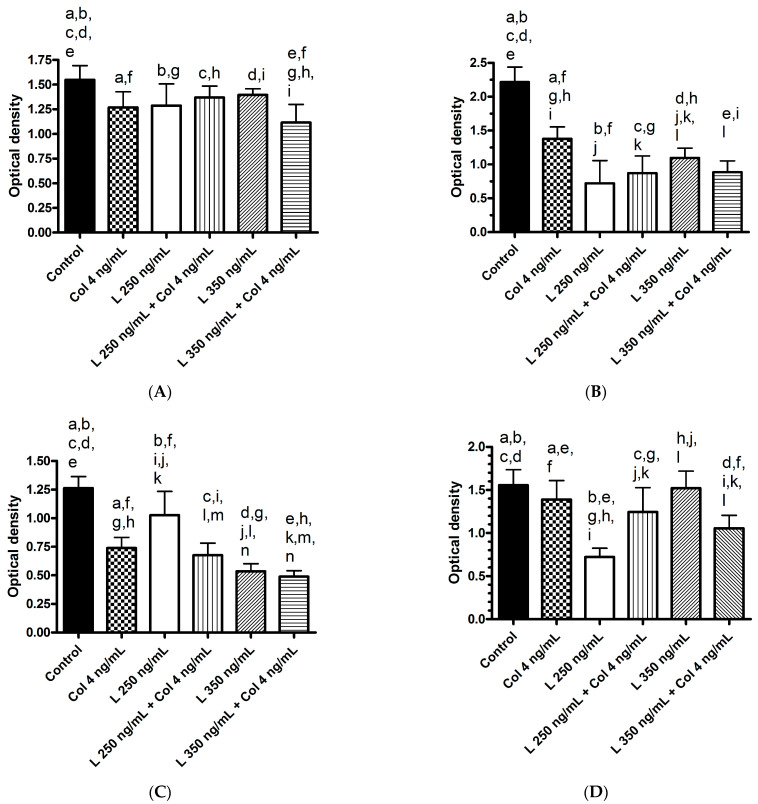
Anti-proliferative effects of colchicine, Lenvatinib, or their combinations on four primary cultured hepatocellular carcinoma cells. The results from 16 replicated wells were calculated to obtain the mean and standard deviation (SD) for unpaired two-tailed *t*-test analysis. Significant difference between two means were marked with the same lowercase letters. Bars indicate SD. (**A**) S103 cells, *p* values: a, e, h < 0.0001; b = 0.0004; c = 0.0005; d = 0.042; f = 0.0174; g = 0.0233; i = 0.0011, (**B**) S143 cells, *p* values: a, b, c, d, e, f, g, h, i < 0.0001; j = 0.0003; k = 0.0044; l = 0.0006, (**C**) S160 cells, *p* values: a, c, d, e, f, g, h, i, j, k, l, m < 0.0001; b = 0.0004; n = 0.0348, (**D**) S176 cells, *p* values: a = 0.0263; b, d, e, f, g, h, i, l < 0.0001; c = 0.0009; j = 0.0033; k = 0.0262.

**Table 1 cancers-15-05097-t001:** The effects of colchicine, lenvatinib, or their combinations on differential expressions of tested genes.

Experimental Drug Concentrations	Col 4 ng/mL	L 250 ng/mL	L350 ng/mL	L250 ng/mL + Col 4 ng/mL	L350 ng/mL + Col 4 ng/mL
S103					
Up-regulation	*NANOG* (2.91)	*FGFR1* (1.54), *FGFR3* (2.98), *FLT4* (2.7), *KDR* (2.51), *KIT* (1.88), *NANOG* (3.94), *PDGFRB* (1.86)	*FGFR1* (1.61), *FGFR3* (1.84), *FLT4* (2.24), *KDR* (2.3), *KIT* (1.9), *NANOG* (3.82), *PDGFRA* (1.42), *PDGFRB* (1.74)	*FGFR1* (1.8), *FGFR3* (2.41), *FLT4* (2.23), *KIT* (1.59), *NANOG* (1.88)	*FGFR1* (1.59), *FGFR3* (1.48), *FLT4* (1.63), *KIT* (1.75), *NANOG* (1.8)
Down-regulation	*FLT1* (0.64), *KIT* (0.62), *PDGFRA* (0.67)	*FGFR4* (0.28), *FLT1* (0.08)	*FGFR4* (0.27), *FLT1* (0.03)	*FGFR2* (0.47), *FGFR4* (0.16), *FLT1* (0.05), *RET* (0.41)	*FGFR2* (0.47), *FGFR4* (0.14), *FLT1* (0.01), *PDGFRA* (0.45), *RET* (0.2)
S143					
Up-regulation	*FGFR1* (1,39)	*FGFR1* (1.53), *FGFR2* (1.74), *FGFR3* (13.52), *FGFR4* (2.53), *FLT4* (3.06), *KIT* (1.71), *NANOG* (1.34), *PDGFRA* (1.47), *PDGFRB* (8.36)	*FGFR3* (5.71), *NANOG* (2.0), *PDGFRB* (2.22)	*FGFR1* (1.47), *FGFR3* (1.73),	*FGFR1* (1.82), *FGFR3* (4.75), *FGFR4* (2.01), *FLT4* (1.39), *NANOG* (1.47), *PDGFRB* (1.79)
Down-regulation	*KDR* (0.41), *NANOG* (0.65), *PDGFRA* (0.33), *PDGFRB* (0.69), *RET* (0.61)	*RET* (0.59)	*FLT1* (0.46), *RET* (0.17)	*FLT1* (0.5), *FLT4* (0.58), *KDR* (0.32), *PDGFRA* (0.28)	*FLT1* (0.25), *KDR* (0.33), *PDGFRA* (0.28)
S160					
Up-regulation		*FLT4* (1.32), *NANOG* (1.37)	*FGFR3* (2.88), *NANOG* (1.47)		
Down-regulation	*FLT1* (0.45), *FLT4* (0.63), *FGFR2* (0.4), *FGFR3* (0.29), *FGFR4* (0.47), *KDR* (0.67), *KIT* (0.03), *NANOG* (0.5), *PDGFRA* (0.11), *PDGFRB* (0.43), *RET* (0.2)	*FGFR1* (0.65), *FGFR2* (0.48), *FLT1* (0.55), *KIT* (0.07), *PDGFRA* (0.18)	*FGFR1* (0.66), *FGFR2* (0.31), *FLT1* (0.46), *KIT* (0.15), *PDGFRA* (0.09), *RET* (0.08)	*FGFR2* (0.35), *FGFR3* (0.61), *FGFR4* (0.48), *KDR* (0.61), *KIT* (0.14), *PDGFRA* (0.02), *PDGFRB* (0.28), *RET* (0.08)	*FGFR2* (0.35), *FGFR3* (0.54), *FGFR4* (0.51), *FLT1* (0.67), *KDR* (0.39), *KIT* (0.08), *NANOG* (0.50), *PDGFRA* (0.02), *PDGFRB* (0.43), *RET* (0.41)
S176 *					
Up-regulation	*FGFR1* (2.49), *KIT* (2.61)	*NANOG* (1.79)	*FGFR4* (1.49)	*FGFR1* (2.58), *KIT* (2.31)	*FGFR1* (2.94), *KIT* (2.65)
Down-regulation	*FGFR3* (0.27), *FGFR4* (0.55), *NANOG* (0.46), *PDGFRA* (0.42), *PDGFRB* (0.67), *RET* (0.57)	*FLT1* (0.54), *KIT* (0.54), *PDGFRA* (0.43), *RET* (0.62)	*FLT1* (0.59), *RET* (0.52)	*FGFR3* (0.58), *FGFR4* (0.48), *NANOG* (0.29), *PDGFRA* (0.34), *PDGFRB* (0.59), *RET* (0.45)	*FGFR2* (0.76), *FGFR3* (0.65), *FGFR4* (0.62), *NANOG* (0.15), *PDGFRA* (0.23), *PDGFRB* (0.33), *RET* (0.56)

Eleven lenvatinib target genes (*FGFR1*, *FGFR2*, *FGFR3*, *FGFR4*, *FLT1*, *FLT4*, *KDR*, KIT, *PDGFRA*, *PDGFRB*, *RET*) and one gene (*NANOG*) of a cancer stem cell marker were included for investigation. Each gene received triple quantitative reverse transcriptase-polymerase chain reactions to obtain the mean value. The mean magnitude of either up-regulated or down-regulated gene expression fold change was shown within the parentheses. Up-regulation of gene was defined as gene expression fold change > 1.3 and down-regulation of gene was defined as gene expression fold change < 0.7. *: S176 did not show detectable expressions of *FLT4* and *KDR*. Col: colchicine; L: lenvatinib.

**Table 2 cancers-15-05097-t002:** The number of lenvatinib target genes with significantly differential expressions caused by colchicine, Lenvatinib, or their combinations.

Drug Concentrations	Col 4 ng/mL	L 250 ng/mL	L350 ng/mL	L250 ng/mL + Col 4 ng/mL	L350 ng/mL + Col 4 ng/mL
S103					
Up-regulation	0	6	7	4	4
Down-regulation	3	2	2	4	5
S143					
Up-regulation	1	8	2	2	5
Down-regulation	4	1	2	4	3
S160					
Up-regulation	0	1	1	0	0
Down-regulation	10	5	6	8	9
S176					
Up-regulation	2	0	1	2	2
Down-regulation	5	4	2	5	6

Eleven lenvatinib target genes (*FGFR1*, *FGFR2*, *FGFR3*, *FGFR4*, *FLT1*, *FLT4*, *KDR*, KIT, *PDGFRA*, *PDGFRB*, *RET*) were studied with quantitative reverse transcriptase-polymerase chain reaction. Col: colchicine; L: lenvatinib.

**Table 3 cancers-15-05097-t003:** Summary of the anti-cancer effects of colchicine, lenvatinib, and their combinations on primary cultured hepatocellular carcinoma cell lines.

Cell Lines	S103	S143	S160	S176 *
(a) Strength of significant anti-proliferative effect	L350 + Col > (Col, L250, L250 + Col, L350)	(L250, L250 + Col, L350 + Col) > L350 > Col	L350 + Col > L350 > (Col, L250 + Col) > L250	L250 > L350 + Col > (Col, L250 + Col)
(b) Number of lenvatinib target genes				
Up-regulation	L350 > L250 > (L250 + Col, L350 + Col)	L250 > L350 + Col > (L250 + Col, L350) > Col	(L250, L350)	(L250 + Col, L350 + Col, Col) > L350
Down-regulation	L350 + Col > L250 + Col > Col > (L250, L350)	(Col, L250 + Col) > L350 + Col > L350 > L250	Col > L350 + Col > L250 + Col > L350 > L250	L350 + Col > (L250 + Col, Col) > L250 > L350
(c) Magnitude of *NANOG* expression				
Up-regulation	L250 > L350 > Col > (L250 + Col, L350 + Col)	L350 > L350 + Col > L250	L350 > L250	L250
Down-regulation		Col	(L350 + Col, Col)	L350 + Col > L250 + Col > Col

Parentheses indicate similar effects. *: There was no significant anti-proliferative effect caused by L350 as compared with control in S176. Col: colchicine 4 ng/mL, L250: 250 ng/mL lenvatinib, L350: 350 ng/mL lenvatinib.

## Data Availability

The data that support the findings of this study are available from the corresponding author, upon reasonable request.

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
