# Peer review of "Dose Consideration of Lenvatinib’s Anti-Cancer Effect on Hepatocellular Carcinoma and the Potential Benefit of Combined Colchicine Therapy"

_cancers, 2023, doi:10.3390/cancers15205097_

Round 1

Reviewer 1 Report

This manuscript delves into the anti-cancer properties of lenvatinib, a multi-kinase inhibitor, in hepatocellular carcinoma (HCC) cell lines derived from patient samples. In a surprising departure from conventional wisdom regarding dose-dependent anti-cancer effects, the study reveals that lenvatinib does not consistently adhere to this pattern in HCC cells. Instead, it predominantly impacts specific genes associated with angiogenesis and proliferation, rather than directly inducing cytotoxicity. Moreover, the research underscores the heterogeneous nature of HCC cells, including the presence of cancer stem cells expressing the NANOG gene. Paradoxically, lenvatinib treatment up-regulates NANOG in various HCC cell lines, potentially bolstering cancer stem cell characteristics and facilitating tumor progression. The authors suggest combining lenvatinib with colchicine, a potent microtubule destabilizer. Colchicine not only exerts direct cytotoxic effects but also exerts a more pronounced influence on lenvatinib target genes compared to lenvatinib alone. Furthermore, it counteracts lenvatinib-induced NANOG up-regulation. In summary, this study challenges the notion of lenvatinib's dose-dependent anti-cancer effects in HCC and proposes a promising strategy of combining lenvatinib with colchicine to enhance the overall anti-cancer response. However, key questions remain:

1. To gain a deeper understanding of lenvatinib's dose-dependent effects, the authors should extend the dose-response curve analysis to cover a broader range of concentrations, not only two dosage (250 ng/mL and 350 ng/mL in this study). This expansion may elucidate a concentration threshold beyond which lenvatinib's impact levels off. Furthermore, exploring the time-dependent effects of lenvatinib on HCC cells by varying treatment durations could offer valuable insights into the kinetics of gene expression alterations and anti-proliferative mechanisms.

2. It would be beneficial for the authors to provide a clear rationale for including NANOG in the gene expression experiments. Additionally, they should address the choice of other cancer stem cell markers or justify the focus on NANOG. The authors should also elucidate the reasoning behind combining colchicine with lenvatinib and whether there is a specific scientific basis for investigating the eleven lenvatinib target genes, clarifying whether these genes are considered key or hub genes in this context.

Minor:

1. An explanation regarding the observation of higher lenvatinib concentrations having less cytotoxic effects on certain cell lines than lower concentrations, as seen in S103 and S176, would enhance the manuscript. Furthermore, any insight into why combination treatments exhibit antagonistic effects in specific cell lines, such as L 250 + C combined treatment in S103 and S176, should be provided.

2. The manuscript should aim to determine the optimal concentration of colchicine for combination therapy with lenvatinib. Investigating whether lower or higher colchicine concentrations yield superior anti-cancer effects.

3. If feasible, the authors should consider conducting RNA sequencing (RNA-Seq) analysis to obtain a comprehensive transcriptome profile of HCC cells treated with lenvatinib, colchicine, or their combination. This approach can unveil additional genes and pathways affected by these treatments.

4. Exploring potential correlations between the tumor stages of patients from whom the cells were derived and the response to colchicine and lenvatinib could provide valuable clinical insights.

5. The use of a Venn Diagram to visually depict the overlap and distinctions in the differential expressions of tested genes under different treatment conditions would enhance the clarity of the results.

6. In addition to investigating anti-proliferation, the authors could conduct other functional experiments to assess the broader anti-tumor effects of lenvatinib, providing a more comprehensive evaluation of its therapeutic potential.

Author Response

We greatly appreciate your kindness to give us an opportunity to revise this manuscript. Our replies to the comments are described in the following. We sincerely hope that you can accept our explanations and revisions.

  1. To gain a deeper understanding of lenvatinib's dose-dependent effects, the authors should extend the dose-response curve analysis to cover a broader range of concentrations, not only two dosage (250 ng/mL and 350 ng/mL in this study). This expansion may elucidate a concentration threshold beyond which lenvatinib's impact levels off. Furthermore, exploring the time-dependent effects of lenvatinib on HCC cells by varying treatment durations could offer valuable insights into the kinetics of gene expression alterations and anti-proliferative mechanisms.

Reply:

We appreciate the reviewer’s this very valuable comment. However, we are very sorry that we are unable to perform further experiments using broader range of concentrations because these cell lines were stored at their 8th or more passage. Further experiments using these stored cells may have bias in comparison with the present results and are hard to be applied in clinical situation. This is due to that repeated in vitro cultures can cause homogeneity of cells and alterations in gene expressions as shown in our previous report (Biomed. Pharmacother. 66(2012)454–458). Moreover, the present results in combination with our previous experiments on sorafenib and regorafenib (reference 16) indicated that the responses of HCC to these drugs were determined by characters of cancer cells. Therefore, the possible concentration threshold and the time-dependent effects of lenvatinib for one cell line cannot be correctly apply to other cell lines. We revised the sentences in section of discussion last paragraph (line 284-292).

: Since different characters of cancer cells determine the interaction of genes and signaling pathways and the composition of HCC tumor is heterogeneity, the actually clinical significance for those up-regulated genes induced by colchicine, lenvatinib or their combinations still need to be further investigated. On the other hand, the present study was unable to elucidate the definite relation between cellular character and lenvatinib effects on HCC due to only four cell lines studied. Further large-scale extensive experiments are required to clarify this important topic. Nevertheless, the present results can provide the new insight for the understanding the effects of lenvatinib on HCC and the potential role of colchicine on combined therapy.

We also revised the conclusion to describe the potential clinical applications of our results (line 297-301).

: Lenvatinib does not show dose-dependent anti-cancer effect on HCC. In combined consideration of lenvatinib side effects and our results, starting from low rather than high therapeutic dose and gradually increasing dose in case of no obvious evidence of anti-cancer effects might be a suitable way to prescribe this drug. Combined colchicine with lenvatinib can promote the total anti-cancer effects on HCC.

  1. It would be beneficial for the authors to provide a clear rationale for including NANOG in the gene expression experiments. Additionally, they should address the choice of other cancer stem cell markers or justify the focus on NANOG. The authors should also elucidate the reasoning behind combining colchicine with lenvatinib and whether there is a specific scientific basis for investigating the eleven lenvatinib target genes, clarifying whether these genes are considered key or hub genes in this context.

Reply:

We appreciate the reviewer’s this very valuable comment.

  • The rationale for including NANOG in the gene expression experiments was described in section of introduction. We revised the following sentence (line 70-75).

Original: Since the cancer stem cell marker NANOG had been shown as a hazard factor to predict poor prognosis in patients with HCC [15], NANOG in combination with eleven lenvatinib target genes were applied for gene expression experiments.

Revised:

: NANOG is a cancer stem cell marker which had been shown as a hazard factor to predict poor prognosis in patients with HCC [15]. Our previous experiments using two other multikinase molecular inhibitors including sorafenib and regorafenib in primary cultured HCC cells showed that these drugs could up-regulated NANOG within their therapeutic ranges [16]. Therefore, NANOG was selected as one of the target genes.

The reasons to select eleven lenvatinib target genes for experiments were based on previous reports (reference 2-4). The following sentence was added in section of introduction (line 75-77).

: Eleven genes including FGFR1, FGFR2, FGFR3, FGFR4, FLT1 (VEGFR1), FLT4 (VEGFR3), KDR (VEGFR2), KIT, PDGFRA, PDGFRB, RET were selected as lenvatinib target genes based on the previous reports [2-4].

Minor:

  1. An explanation regarding the observation of higher lenvatinib concentrations having less cytotoxic effects on certain cell lines than lower concentrations, as seen in S103 and S176, would enhance the manuscript. Furthermore, any insight into why combination treatments exhibit antagonistic effects in specific cell lines, such as L 250 + C combined treatment in S103 and S176, should be provided.

Reply:

We appreciate the reviewer’s this very valuable comment. The only explanation for these results was that the intra-cellular gene interaction and signaling pathway are determined by the characters of cancer cells as described in section of discussion. This was one of the key points in the manuscript to show the important impact of cellular characters on the determination the lenvatinib anti-cancer effects on HCC. However, the present study was unable to elucidate the definite relation between cellular character and the effects of lenvatinib on HCC due to only four cell lines studied. Further large-scale extensive experiments are required to resolve this important topic. We add the following sentence in section of discussion last paragraph (line 284-286).

: On the other hand, the present study was unable to elucidate the definite relation between cellular character and lenvatinib effects on HCC due to only four cell lines studied. Further large-scale extensive experiments are required to clarify this important topic.

  1. The manuscript should aim to determine the optimal concentration of colchicine for combination therapy with lenvatinib. Investigating whether lower or higher colchicine concentrations yield superior anti-cancer effects.

Reply:

We appreciate the reviewer’s this very valuable comment. The dose-dependent anti-cancer effect of colchicine on HCC had been report in our previous study (Life Sci. 2013, 93, 323–328. reference 13) using clinically feasible colchicine concentrations of 2 ng/mL and 6 ng/mL (reference 16-18). Therefore, we selected 4 ng/mL for this experiment. We sincerely hope that the reviewer can accept our explanation.

  1. If feasible, the authors should consider conducting RNA sequencing (RNA-Seq) analysis to obtain a comprehensive transcriptome profile of HCC cells treated with lenvatinib, colchicine, or their combination. This approach can unveil additional genes and pathways affected by these treatments.

Reply:

We appreciate the reviewer’s this very valuable comment. However, we are very sorry that we are unable to perform further experiments due to the reason described in reply for the comment one. Moreover, the results from only four cell lines might be very hard to obtain a conclusive result due to that the responses of HCC to lenvatinib are influenced by the different characters of cancer cells. We sincerely hope that the reviewer can accept our explanation.

  1. Exploring potential correlations between the tumor stages of patients from whom the cells were derived and the response to colchicine and lenvatinib could provide valuable clinical insights.

Reply:

We appreciate the reviewer’s this very valuable comment. However, the results from only four cell lines were very hard to obtain a conclusive result. We sincerely hope that the reviewer can accept our explanation.

  1. The use of a Venn Diagram to visually depict the overlap and distinctions in the differential expressions of tested genes under different treatment conditions would enhance the clarity of the results.

Reply:

We appreciate the reviewer’s this very valuable comment. However, we are very sorry that we are unable to present the results by suitable Venn Diagram. We sincerely hope that the reviewer can accept our explanation.

  1. In addition to investigating anti-proliferation, the authors could conduct other functional experiments to assess the broader anti-tumor effects of lenvatinib, providing a more comprehensive evaluation of its therapeutic potential.

Reply:

We appreciate the reviewer’s this very valuable comment. However, we are very sorry that we are unable to perform further experiments due to the reason described in reply to the comment one. We sincerely hope that the reviewer can accept our explanation.

Reviewer 2 Report

The results of several recent studies performed in humans with advanced hepatocellular carcinoma (HCC)  have represented a great advance in the management of this disease, and the authors conclude that Sorafenib is, to date, the only systemic treatment that has demonstrated an increase in survival in HCC and, after its approval by the FDA and the EMEA, It is currently the reference treatment for advanced HCC.

Likewise, these positive results constitute proof that molecular therapies are effective in HCC and open the door for the investigation of new agents, as single agents or in combination with sorafenib, and for the evaluation of sorafenib in other settings, such as in adjuvant therapy after the application of potentially curative treatments such as ablation or surgical resection, or associated with chemoembolization in patients with intermediate HCC. 

S. Wilhelm, C. Carter, M. Lynch, T. Lowinger, J. Dumas, R.A. Smith, et al., Discovery and development of sorafenib: a multikinase inhibitor for treating cancer. Nat Rev Drug Discov, 5 (2006), pp. 835-844. http://dx.doi.org/10.1038/nrd2130

 G.K. Abou-Alfa, L. Schwartz, S. Ricci, D. Amadori, A. Santoro, A. Figer, et al., Phase II study of sorafenib in patients with advanced hepatocellular carcinoma. J Clin Oncol, 24 (2006), pp. 4293-4300. http://dx.doi.org/10.1200/JCO.2005.01.3441

M. Reig, J. Bruix., Sorafenib for hepatocellular carcinoma: global validation. Gastroenterology, 137 (2009), pp. 1171-1173. http://dx.doi.org/10.1053/j.gastro.2009.07.008

In consequence it would be highly recommended to continue performing experimental or celular studies including sorafenib instead Lenvatinib. It does not seem necessary cochicine in these experiments.

 I suggest to authors to include some comments in this way.

Author Response

We greatly appreciate your kindness to give us an opportunity to revise this manuscript. Our replies to the comments are described in the following. We sincerely hope that you can accept our explanations.

The results of several recent studies performed in humans with advanced hepatocellular carcinoma (HCC) have represented a great advance in the management of this disease, and the authors conclude that Sorafenib is, to date, the only systemic treatment that has demonstrated an increase in survival in HCC and, after its approval by the FDA and the EMEA, it is currently the reference treatment for advanced HCC.

Likewise, these positive results constitute proof that molecular therapies are effective in HCC and open the door for the investigation of new agents, as single agents or in combination with sorafenib, and for the evaluation of sorafenib in other settings, such as in adjuvant therapy after the application of potentially curative treatments such as ablation or surgical resection, or associated with chemoembolization in patients with intermediate HCC. 

  1. Wilhelm, C. Carter, M. Lynch, T. Lowinger, J. Dumas, R.A. Smith, et al., Discovery and development of sorafenib: a multikinase inhibitor for treating cancer. Nat Rev Drug Discov, 5 (2006), pp. 835-844. http://dx.doi.org/10.1038/nrd2130

 G.K. Abou-Alfa, L. Schwartz, S. Ricci, D. Amadori, A. Santoro, A. Figer, et al., Phase II study of sorafenib in patients with advanced hepatocellular carcinoma. J Clin Oncol, 24 (2006), pp. 4293-4300. http://dx.doi.org/10.1200/JCO.2005.01.3441

  1. Reig, J. Bruix., Sorafenib for hepatocellular carcinoma: global validation. Gastroenterology, 137 (2009), pp. 1171-1173. http://dx.doi.org/10.1053/j.gastro.2009.07.008

In consequence it would be highly recommended to continue performing experimental or celular studies including sorafenib instead Lenvatinib. It does not seem necessary cochicine in these experiments.

 I suggest to authors to include some comments in this way.

Reply:

We greatly appreciate reviewer’s this valuable comment. Experiments for sorafenib and regorafenib had been reported in our previous study (Biomed. Pharmacother. 2022, 153, 113540, reference 16). We described this part in section of discussion first paragraph for the comparison with lenvatinib results. The reasons to include colchicine in this study were described in section of introduction (line 56-61, line 64-67, line 72-74). The key points were as following:

  • The long-term benefits of lenvatinib on the treatment of advanced HCC are unsatisfactory and searching a new drug to enhance the lenvatinib effects on HCC is a clinically urgent issue.
  • Colchicine within its clinically achievable plasma concentrations had been shown to have dose-dependent anti-cancer effect on primary cultured HCC cells and cancer-associated fibroblasts (reference 13). This drug had also been shown to have the potential for the palliative treatment of advanced HCC in our phase IIa clinical trial (reference 14).
  • Our previous study (reference 16) showed that colchicine can obliterated or decreased the magnitude of up-regulated NANOG induced by sorafenib or regorafenib.

Reviewer 3 Report

The Authors present the results of a good experimental study with an interesting mechanicistic rationale. 

Some comments:

- the Authors should greatly expand on the potential significance of the differentially-expressed genes identified in the study.

- I understand that English is not Authors' native language, but they should make a major effort to improve language and style.

- can the Authors expand on how they foresee the potential clinical applications of their findings?

See above

Author Response

We greatly appreciate your kindness to give us an opportunity to revise this manuscript. Our replies to the comments are described in the following. We sincerely hope that you can accept our explanations and revisions.

The Authors present the results of a good experimental study with an interesting mechanicistic rationale. 

Some comments:

- the Authors should greatly expand on the potential significance of the differentially-expressed genes identified in the study.

Reply:

We greatly appreciate reviewer’s this valuable comment. The genes selected for experiments included eleven reported lenvatinib target genes elated to angiogenesis and proliferation (reference 2-4) and one gene (NANOG) for cancer stem cell marker. The clinical significance of up-regulated NANOG was discussed in section of discussion first paragraph (line 253-259). The anti-cancer effects of lenvatinib were down-regulation of its target genes to inhibit cancer progression. However, the present study showed that lenvatinib could reversely up-regulate various target genes among different cell lines caused by different lenvatinib concentrations. Since the intra-cellular interaction of genes and signaling pathways were very complex and determined by the characters of cancer cells, the clinical significance of up-regulated lenvatinib target genes identified in the study still needs to be further clarified. Nevertheless, the present results can provide the new insight for the understanding the effects of lenvatinib on HCC and the potential role of colchicine on combined therapy.

The following sentence was added in section of discussion (last paragraph, line 280-283).

: Since different characters of cancer cells determine the interaction of genes and signaling pathways and the composition of HCC tumor is heterogeneity, the actually clinical significance for those up-regulated genes induced by colchicine, lenvatinib or their combinations still need to be further investigated.

- I understand that English is not Authors' native language, but they should make a major effort to improve language and style.

Reply:

We do our best to revise the manuscript.

- can the Authors expand on how they foresee the potential clinical applications of their findings?

Reply:

We greatly appreciate reviewer’s this valuable comment. We revised the sentences in section of conclusion (line 293-297).

: Lenvatinib does not show dose-dependent anti-cancer effect on HCC. In combined consideration of lenvatinib side effects and our results, starting from low rather than high therapeutic dose and gradually increasing dose in case of no obvious evidence of anti-cancer effects might be a suitable way to prescribe this drug. Combined colchicine with lenvatinib can promote the total anti-cancer effects on HCC.

Round 2

Reviewer 1 Report

The authors have addressed all my concerns and I have no further questions.